# Upconversion Nanomaterials in Bioimaging and Biosensor Applications and Their Biological Response

**DOI:** 10.3390/nano12193470

**Published:** 2022-10-04

**Authors:** Zayakhuu Gerelkhuu, Yong-Ill Lee, Tae Hyun Yoon

**Affiliations:** 1Department of Chemistry, College of Natural Sciences, Hanyang University, Seoul 04763, Korea; 2Institute of Next Generation Material Design, Hanyang University, Seoul 04763, Korea; 3Department of Materials Convergence and System Engineering, Changwon National University, Changwon 51140, Korea; 4Faculty of Chemical Engineering, Industrial University of Ho Chi Minh City, Ho Chi Minh City 71408, Vietnam; 5Research Institute for Convergence of Basic Science, Hanyang University, Seoul 04763, Korea

**Keywords:** lanthanide-based nanomaterials, upconversion nanomaterials, bioimaging, biosensor, biological response

## Abstract

In recent decades, upconversion nanomaterials (UCNMs) have attracted considerable research interest because of their unique optical properties, such as large anti-Stokes shifts, sharp emissions, non-photobleaching, and long lifetime. These unique properties make them ideal candidates for unified applications in biomedical fields, including drug delivery, bioimaging, biosensing, and photodynamic therapy for specific cancers. This review describes the general mechanisms of upconversion, synthesis methods, and potential applications in biology and their biological responses. Additionally, the biological toxicity of UCNMs is explained and summarized with the associated intracellular association mechanisms. Finally, the prospects and future challenges of UCNMs at the clinical level in biological applications are described, along with a summary of opportunity for biological as well as clinical applications of UCNMs.

## 1. Introduction

Nanoscience has grown rapidly over the last decade. Recently, nanoscientists have increasingly focused on biological applications. Over the last decade, the development of upconversion nanomaterials (UCNMs) has facilitated the transformation of fluorescence imaging from microscopic to macroscopic imaging. 

Although conventional fluorescent materials, such as organic dyes [1,2], fluorescent proteins [3], metal complexes [4,5], and semiconductor quantum dot-based materials [6,7], have led to significant advances in real-time detection and bioimaging, they still have some limitations. Fluorescent materials, for example, are typically excited by ultraviolet (UV) or visible light (VIS), which can cause autofluorescence in biological samples, DNA damage, and cell death, thus resulting in a low signal-to-noise ratio and limited sensitivity. Furthermore, owing to their broad emission spectra, these fluorescent materials are unsuitable for multiplex biolabeling. Additionally, they frequently exhibit low photostability when exposed to external illumination. Downconversion nanomaterials with a high quantum yield, narrow emission bandwidth, substantial Stokes shift, size-dependent adjustable emission, and strong photostability are appealing alternative luminescent labels for imaging [7,8,9]. However, the possible toxicity of down-conversion nanomaterials limits their biological applications [10]. Therefore, lanthanide-based UCNMs are primarily used in biological applications. 

UCNMs initially attracted attention for applications in the biomedical field because of their anti-Stokes luminescence; they emit at shorter wavelengths and absorb long-wavelength photons via two or more photons. Using long-wavelength photon sources as an excitation light, prevents tissue damages, affords deep tissue-penetrating ability, and can effectively avoid interference from the fluorescence of the living body [11]. Additionally, UCNMs with a large Stokes shift, weak photobleaching, low toxicity, and good stability have excellent potential for biological applications compared to conventional fluorescent materials. Therefore, UCNMs are suitable for various biological and medical applications, such as early diagnosis of disease [12,13,14,15], bioimaging [16,17,18], biosensors [11,19,20,21], and cancer treatment [22,23,24]. 

This review article presents numerous applications for UCNMs, including bioimaging and biosensors, as well as their biological responses and conversion system mechanisms. Furthermore, the proposed intercellular association pathways in the UCMNs are examined. The final section discusses the future perspectives and challenges of nanotoxicity pathways. 

### 1.1. Mechanisms of Upconversion Nanomaterials 

Upconversion materials produce a distinct type of photoluminescence (PL) in which lower-energy excitation is converted to higher-energy emission via a multiphoton absorption process [25,26]. Inorganic host lattices and lanthanide dopant ions are generally used to fabricate upconversion nanomaterials. When immersed in a solid or inorganic host lattice, lanthanides, actinides, and transition metal ions can emit upconversion fluorescence. Several upconversion mechanisms have been proposed, including excited-state absorption (ESA), energy transfer upconversion (ETU), photon avalanche (PA), and cooperative sensitization upconversion (CSU). New processes, such as energy migration-mediated upconversion (EMU), have recently been reported (Figure 1). ESA is the simplest basic upconversion process, with excitation taking the form of a sequence of pump photon absorptions by a single ground-state (G) ion. When an ion is excited from the ground state to E1, a second pump photon promotes the ion from E1 to the higher-lying state E2 and causes UC emission before it decays to the G state. ESA is only essential when the doping concentration is low; otherwise, the energy transfer between the two activators is minimal, resulting in limited absorption and low efficiency.

After absorbing a pump photon, Ion 1, known as the sensitizer, is excited from the G state to its metastable level E1 in the ETU. Then, Ion 1 transfers its harvested energy to the G state and the excited state E1 of Ion 2, known as the activator, exciting Ion 2 to its upper emitting state E2. Meanwhile, the sensitizer Ion 1 relaxes back to the G state. The dopant concentration, which affects the average distance between the surrounding dopant ions, significantly affects the UC efficiency of the ETU process. However, the UC efficiency in an ESA process is independent of the dopant concentration owing to its single-ion characteristic. 

PA-induced UC has a unique pump mechanism that necessitates a pump intensity above a certain threshold value. This process can form a feedback loop with the help of resonant cross-relaxation (CR) between the excited ion and another adjacent G-state ion, which triggers a rapid accumulative effect of the intermediate-level population and leads to strong UC emission as an avalanche process. 

Cooperative sensitization upconversion (CSU) involves the interaction of three ions. The contained energy is collaboratively transferred by two excited neighboring sensitizer ions to one activator ion at a time, and the excited activator relaxes back to its G-state by producing an upconverted photon [27]. Compared to other approaches, CSU has a substantially lower likelihood because the emitters lack an intermediate energy level that is metastable for the sensitizer. 

Sensitizers, accumulators, migrators, and activators are the four types of luminescent centers involved in the energy migration-mediated upconversion (EMU) process [28]. The sensitizer/accumulator and activator are separated by core–shell layers and are linked by migrators. The sensitizer is excited by the G-state absorption first and then transmits its energy to the accumulator, promoting it to a higher excited state. To obtain energy from the sensitizer, the accumulator must have energy levels with longer lives. The energy then migrates from the high-excited-state accumulator to the migrator, followed by the excitation energy migration through the core–shell interface. Finally, the migrated energy is trapped by the activator in the shell and emits upconversion luminescence (UCL).

### 1.2. Synthesis of Upconversion Nanomaterials

The development of synthetic strategies is important for the efficient upconversion nanomaterials with defined size, shape, composition, and phase. The synthesis methods of UCNMs include thermal decomposition, hydrothermal/solvothermal, precipitation/coprecipitation, and sol–gel processes. The thermal decomposition method produces size-controlled and well-shaped nanoparticles [29] within a short reaction time. High-boiling-point organic solvents frequently include the surfactant-assisted decomposition of precursors. At a relatively high temperature (~300 °C), the ions created are joined to form new nuclei. Organic precursors, such as trifluoroacetate and oleate compounds, as well as polar capping groups, including oleic acid [30], oleylamine [31], and octadecene as surfactants [32], are often utilized. Solvents are thought to regulate particle development by capping the surface of the UCNMs. 

Due to the benefit of experimental setups with autoclave reactors, solvothermal and hydrothermal techniques require relatively moderate temperatures (<250 °C). Despite the moderate conditions, these approaches allow the creation of UCNMs with a decent level of crystallinity with good control over size and shape by fine-tuning the experimental temperature and reaction time, the nature of the solvent and surfactant, and their molar ratios. In a brief synthesis process, the reaction precursors, solvents, and surfactants containing functional groups were combined and heated in an autoclave. 

The coprecipitation approach was designed to overcome the limitations of the thermal decomposition method and is widely used in the synthesis of UCNMs. This involves the simultaneous precipitation of the two substances. UCNMs are produced with organic surfactants using the coprecipitation method, and their adsorption to the surface of the particles prevents agglomeration [33]. The absence of harmful by-products, the use of low-cost equipment, and easy operation are advantages of the coprecipitation approach. Capping agents such as polyethyleneimine (PEI) [34,35,36], polyvinylpyrrolidone (PVP) [37], polyethylene glycol (PEG) [38], and polyacrylic acid (PAA) [39] can be used to enhance the nanocrystal nucleation.

Thin-films, oxides, and fluoride nanoparticles have been created via a sol–gel process [29]. This approach is a typical wet-chemical process, which begins with a liquid solution of molecular precursors and progresses to forming a new sol phase via hydrolysis and polycondensation. The sol is agglomerated into a gel through a vast molecular network with the addition of a base, followed by annealing at a high temperature for a few hours. Annealing promotes crystallinity and eliminates the solvent from the gel. 

## 2. Physicochemical Properties of Upconversion Nanomaterials and Their Effects on Biological Responses

This section discusses the effect and impact of the physicochemical properties of UCNMs, including the size, shape, surface modification, functional groups, optical properties, and their effects on cellular uptake, cell toxicity, and biological responses, as shown in Figure 2. Table 1 summarizes several recent examples of UCNMs and their corresponding physical and chemical properties, which play significant roles in the cellular and molecular signaling pathways of biological systems. 

### 2.1. Size (Core and Hydrodynamic)

Size is critical in biological–NP interactions. The biological–NP-related mechanisms include pinocytosis, phagocytosis, clathrin-mediated endocytosis, caveolae-mediated endocytosis, clathrin- and caveolae-independent endocytosis, and micropinocytosis [53]. Cellular uptake and particle processing efficiency in the endocytic path depend on the ion release rate, time dependence, and interactions with cell membranes [54]. In general, NP toxicity influences the human biological system owing to its size-dependent ability to enter the cellular system. As the particle size decreased, the cellular association of NPs on the cell surfaces increased with the volume ratio. Thus, NP toxicity or cellular stress increases with the size of the NPs, followed by penetration into the outer or inner surface of cells [54]. Likewise, NPs with a size of less than 50 nm have a more significant impact on all types of tissues after intravenous injection, exerting strong toxic effects [55]. 

NP size refers to their “in vivo” dispersion or pharmacological behavior [56], which directly influences physiological activities. NPs larger than 1 µm do not easily penetrate cells, although they interact with absorbed proteins. The kidney does not excrete NPs larger than 6 nm; therefore, NPs accumulate in certain organs [57]. For example, NPs remain in the tissue and induce hepatoxicity when they come into contact with cadmium selenide quantum dots [58].

Sonavane et al. investigated the biodistribution and bioaccumulation of gold nanoparticles (AuNP) of various sizes in the blood. They discovered that the smaller particles stayed in circulation longer and thus accumulated more in all organs [59].

As shown in the summary in Table 1, the size of UCNMs can range from sub-10 nm to near 100 nm [60]. 

### 2.2. Shape

The shape is a physicochemical characteristic that determines the material toxicity [61]. NPs exist in various forms and configurations, including tubes, fibers, spheres, and planes. Therefore, shape may affect endocytosis, internalization, biodistribution, and removal. For example, spherical nanoparticles of the same size are shown to be more readily and rapidly engulfed by endocytosis than rod-shaped nanoparticles, which can be explained by the longer membrane wrapping time required for elongated particles. Furthermore, spherical particles are less hazardous [62]. As shown in the summary in Table 1, in most cases, the crystal structure of the UCNMs was hexagonal. 

### 2.3. Surface Modification 

The NP surface influences NP–cell interactions and solubility [63]. Changing the chemical, magnetic, optical, and electrical characteristics of NPs might alter their cytotoxic properties by affecting their accumulation, pharmacokinetics, distribution, and toxicity [64].

Surface charges govern an organism’s reaction to changes in NP shape and size in the form of cellular accumulation, which is referred to as colloidal behavior [65]. Surface chemistry influences NP absorption [66], colloidal behavior, plasma protein binding [67], and crossing the blood–brain barrier [68]. The cytotoxicity of NP increases as the surface charge increases [69]. This result implies that more positive charges caused more cell electrostatic interactions, which increased endocytic uptake. Positively charged NPs, however, may be more toxic than negatively charged NPs [70]. Positively charged NP densities may be more readily separated in the interstitial space and thus taken up by tumor cells. Furthermore, positively charged NPs accumulate more than negatively charged NPs in tumors [56].

Surface chemical modification is a key technique in biological applications to reduce toxicity and increase stability, control, and modulate the cellular internalization [71]. Surface functionalization mainly involves polyethylene glycol (PEG), a negative carboxyl group, a neutral hydroxyl, and amine groups. For example, the NP surface can be functionalized with appropriate polymers, such as PEG, to decrease non-specific interactions and achieve selective binding to cell receptors.

Another important factor that influences pharmacokinetics and biodistribution is hydrophobicity. NPs with a more hydrophobic surface absorb plasma proteins, reducing their circulation time [72]. Because of the internal membrane hydrophobicity gap in cells, the surface membrane absorption of hydrophobic C60 agglomerates is thermodynamically preferred over semi-hydrophilic ones, according to computer molecular modeling research [73].

Bio-applications are one of the most important types of applications for UCNMs. The ability to make hydrophobic UCNMs water-soluble and supply reactive groups for subsequent bioconjugation to diverse biomolecules is a critical challenge in the use of hydrophobic UCNMs for bio-applications. Surface modification can link UCNM synthesis and biological applications, which not only improves the photostability of the NPs by providing favorable interfacial properties but also serves as a possible platform for attaching biological molecules and other conjugated materials for a variety of biomedical applications. 

To increase the stability of specific NPs, ligand molecules on the surface can be switched with other molecules, which can endow the particles with new characteristics or usefulness. In this case, the new ligand molecules must have a strong affinity for the inorganic core to swiftly and efficiently replace the original surfactant molecules.

The coordination mechanism of ligand molecules on the surface of UCNMs also plays an important role in biological applications. With two C-O bonds, the carboxyl ends conjugate to Ln^3+^ through a powerful driving force. Naturally, ligand molecules firmly attached to the NPs surface or each other are less likely to detach from the surface of the particles. As a result, the ligand-exchange process is relatively difficult and lengthy. 

One of the ligand exchange methods is “organic ligand-free,” which involves ligand exchange between oleic acid molecules and inorganic ions (H_3_O^+^). This approach is based on removing oleate ligands from the surface of oleate-capped UCNMs produced by thermal breakdown through a simple acid treatment procedure. Furthermore, water-soluble UCNMs permit additional conjunctions with electronegative hydrophilic groups, such as -NH_2_, -OH, -SH, and -COOH, providing exciting potential for the creation of novel UCNMs for biological applications. 

Creating an extra layer on top of oleate-capped UCNMs by depositing amphiphilic molecules is a powerful strategy for making UCNMs water-soluble. The term amphiphilic refers to the presence of both hydrophobic (tail) and hydrophilic (head) parts. 

Through van der Waals forces, the long alkyl chains in UCNMs can intercalate between hydrophobic oleate molecules, producing a bilayer surrounding the particle surface. In addition, the maximum length of the alkyl chain leads to a good surface coating, which is limited by the length of the oleate. The van der Waals force becomes too weak to stabilize the bilayer when the chain length exceeds the oleate length. The hydrophilic head groups of the amphiphiles are oriented toward the solvent, causing the particles to disperse in the aqueous solution. 

### 2.4. Chemical Composition

In general, UCNMs are created by combining various precursors, including an inorganic host matrix, sensitizer ions, and activator ions, with visible emission and UCL. Rare earth elements with inorganic and organic compositions (NaYF_4_, NaGdF_4_, NaErF_4_, NaLuGdF_4_, and Y_2_O_3_) are promising host nanocomposites. 

In addition to these distinguishing features of NPs, their aggregation state must be considered. Aggregation is affected by surface load, material type, and size. Higher NP concentrations result in greater aggregation and lower toxicity [74]. Accordingly, macrophages remove large particles more efficiently and quickly than small particles, which can easily avoid this defense mechanism [75].

### 2.5. Optical Properties of Upconversion Nanomaterials 

UCNMs were created using rare earth elements, Yb^3+^ ions, which have high two-photon absorption, and various lanthanides (Er^3+^, Tm^3+^, Nd^3+^, Eu^3+^, Gd^3+^, etc.) as sensitizers and activators, respectively. The emission colors of UCNMs can be significantly adjusted by varying the concentrations of the sensitizers and activators. The colorful emissions from UCNMs make them attractive materials for biological applications. 

Bioimaging is an important aspect of cancer diagnosis and tumor growth tracking. For example, MRI, CT, and PET methods have been identified as potential diagnostic tools for cancer patient treatment. Several research groups have investigated the use of UCNMs containing various lanthanide ions (Gd^3+^, Yb^3+^, Ho^3+^, and Sm^3+^) as contrast probes for particular characteristics in MRI, CT, and PET methods for biological tissue assessment. 

## 3. Biological Applications of Upconversion Nanomaterials

### 3.1. Bioimaging and Biosensing Application of UCNM

Rare earth-doped UCNMs have been shown to be attractive platforms for bioimaging [49], biosensing [76,77,78], and cancer treatment [24,60,79,80,81]. UCNMs triggered by near-infrared (NIR) light have a high signal-to-noise ratio, good biological tissue penetration [82], and cause no tissue injury [83]. Furthermore, compared to typical luminescent materials, UCNMs exhibit strong optical stability, high chemical stability, and a narrow gap emission [84,85]. Owing to these advantages, UCNMs can be used in various biological applications. At the pre-clinical level, magnetic resonance (MRI) [14,86,87,88], X-ray computed tomography (CT) [89], positron emission tomography (PET) [90], and single-photon emission tomography (SPECT) [88] have all been employed. 

When excited by near-infrared light (808–980 nm), UCNMs display distinct narrow photoluminescence with higher energy (visible light). Because of their distinctive upconversion luminescence (UCL), UCNMs have the potential to be bioimaging probes with appealing properties, such as minimal auto-fluorescence from biological materials and deep penetration depth [91,92]. Consequently, UCNMs have emerged as innovative small-animal imaging agents. Traditional fluorescent materials have limitations that make them unsuitable for high-contrast in vivo imaging [93]. For example, synthetic organic dyes are rapidly photobleached, rendering them unsuitable for longitudinal tracking tests [94]. Common binary quantum dots contain highly toxic components that may hamper their clinical application. 

UCNMs have also been used in biosensing applications for selectively and sensitively detecting small molecules, such as important ions in cellular metabolism and iron ions in HeLa cells [78]. 

Several initiatives have recently been developed to improve the in vivo and in vitro bioimaging applications. Zhao et al. described azobenzene-modified NaGdF_4_:Y,Yb,Tm@NaGdF_4_ UCNMs and b-cyclodextrin (b-CD)-modified downconversion nanoprobes [95]. In vivo imaging showed that UCNMs convert NIR light to the UV-VIS range, causing the isomerization of azobenzene (Azo) between trans- and cis- isomers and the disassembly of Azo and b-CD. Furthermore, with the 808 nm laser, it may emit in the second near-infrared window, which is promising for in vivo fluorescence imaging owing to the high-resolution imaging with deep tissue penetration and minimal tissue autofluorescence. The originality of these studies is that the K_out_ of the nanoprobes accumulated in the tumor resulted in a four-fold accumulation of probes, offering stable cancer retention for up to 5 h by organizing the probes into larger assembled clusters. 

Przybylska et al. [96] reported the structural and spectroscopic features of Ho^3+^-, Er^3+^-, and Tm^3+^-lanthanide (Ln^3+^)-doped SrF_2_:Yb^3+^ and Ln^3+^ nanoparticles. Hydrothermal synthesis was used to create UCNMs chelated with either trisodium citrate or ammonium citrate tribasic surfactants. An alternative synthesis process for the most common, trisodium citrate-based, was devised. The effects of time and precipitation agent quality on the particle size, shape, agglomeration, and spectroscopic properties were also examined. Because of the potential biological applications of the produced NPs, their cytotoxicity was investigated. The produced NPs were also investigated as possible markers. 

NPs may affect cell function by disrupting plasma membrane integrity, interfering with organelle function, or damaging the cytoskeleton [97]. Cytotoxicity is affected by various factors, including size, shape, functionalization, coating, and cell line employed in the experiment [98]. Therefore, it is critical to examine the toxicity of freshly created NPs, which is one of the goals of this study. Examination of toxicity separates our study from other studies. Confocal imaging was used to study the cellular uptake of nanostructures. This paper provides a detailed report on the optimization of the synthesis technique as well as a comparison of luminescence qualities between products generated under different circumstances and with three distinct emitting lanthanide ions. In addition, the cytotoxicity of the produced products was investigated for both bare and functionalized NPs. 

### 3.2. Advantage of UCNMs for Biological Applications Compared with Other Nanomaterials

Near-infrared (NIR) light photons (long-wavelength) are continually absorbed by UCNMs, which then emit UV-VIS light (short-wavelength) via two or multiphoton processes [92,99]. Compared with conventional organic fluorescent dyes (OFDs), UCNMs have various advantages for use in biological systems. First, certain enzymes and nucleic acids in human tissue absorb UV radiation, causing tissue damage and degeneration, especially in in vivo systems. The excitation wavelength of UCNMs, on the other hand, is generally in the NIR range, and they absorb NIR (usually 808 nm or 980 nm) light. Therefore, using NIR light as an excitation light source not only protects normal tissues but also achieves deep tissue penetration and can effectively avoid interference from the fluorescence of the live organism itself. Second, UCNMs with a significant Stokes shift, low toxicity, and good stability have a great potential for use as integrated probes in diagnosis and therapy. Finally, the emission wavelength of UCNMs may be modulated from the UV-VIS and NIR ranges [80]. UCNMs’ unique features allow them to compound with other nanomaterials such as gold nanoparticles (AuNPs), carbon dots (CDs), fluorescent dyes (FDs), photosensitizers (PSs), graphene, and graphene oxide [100,101]. Fluorescence resonance energy transfer (FRET) allows UCNMs to absorb the emission from these nanomaterials, allowing them to be utilized in related domains. As a result of these advantages, UCNMs have a broad application potential in biomedicine [102], notably for the early diagnosis and cancer treatment [103,104], as evidenced in multiple studies. 

## 4. Biological Response of Upconversion Nanomaterials

UCNMs have been used in immunotherapy for synergistic therapy of tumors. Thus, investigating the biological responses of UCNMs is important for their clinical application. In this section, the mechanism of interaction between UCNMs and biological systems (cells and tissues) was first introduced. Some studies on the toxicity of UCNMs were summarized.

### 4.1. Mechanisms of the Cellular and Tissue Transport of UCNMs

Previous research has shown that the cellular uptake of UCNMs is influenced by NP–cell interactions via several receptor-mediated endocytosis processes [105]. Endocytosis can be classified into numerous categories based on the method, size, and type of cargo. Endocytosis can be classified into four types: (i) phagocytosis, (ii) pinocytosis, (iii) clathrin-mediated endocytosis, and (iv) caveolae-mediated endocytosis, as shown in Figure 3. Endocytosis occurs via a receptor-mediated mechanism, or a non-specific process called pinocytosis, in which the fluid surrounding the cell is taken up regardless of its content. 

Clathrin-mediated endocytosis occurs in specific plasma membrane areas where clathrin is recruited [106]. The engagement of an agonist with its receptor causes the formation of endocytic clathrin-coated vesicles with a size range of 70–150 nm, depending on the cell type, which leads to the assembly of clathrin into a polygonal shape, coating the vesicle. The vesicle then internalizes, losing its clathrin coat, and unites with other vesicles to create an early endosome, which merges with a lysosome.

Caveolin-mediated endocytosis is characterized by plasma membrane invaginations of 60–80 nm in size that may take up extracellular fluid content [53]. The proteins involved in this endocytic pathway, such as caveolin-1, bind to cholesterol in lipid rafts and do not dissolve from vesicles following absorption, as is the case with clathrin-mediated endocytosis. Caveolin vesicles develop and merge with other caveolin vesicles, resulting in multicaveolar structures, known as caveosomes, that fuse bi-directionally with early endosomes. Depending on the cell type, vesicular structures can enter the smooth endoplasmic reticulum or the Golgi-trans network.

Before the internalization, molecules involved in receptor-mediated endocytosis were identified with great specificity. Endocytosis mediated by clathrin and caveolae has been found to mediate folate-coated polymeric NPs with a size of 50–250 nm. The internalization of large particulate materials, such as dust particles and pathogens, is characterized by phagocytosis. Endocytosis of this type is performed by specific cell types and is an important process in innate immunity.

### 4.2. Toxicity Studies of UCNMs

Many studies have investigated the in vitro and in vivo toxicity of UCNMs. Here, we summarize the most up-to-date studies to provide a comprehensive review of the possible adverse effects of UCNMs. Zhou et al. used mouse models and found there was no overt toxicity of NaYF_4_:Yb,Er@SiO_2_ even with a high dose (100 mg/kg) and long-term administration (14 days) [40]. Many in vitro studies have exposed different cell lines to various UCNMs [41,42,43,44,45,46,48,49]. The experimental conditions and results are summarized in Table 1. In most cases, the cell viability was higher than 50%, suggesting the relatively low cytotoxicity of the UCNMs. 

However, more in-depth studies are required to ensure the safety of UCNMs before they enter the market, which will be discussed in the following summary and perspective.

## 5. Summary and Perspective

Over the past few decades, UCNMs significantly improved cancer treatment, deep tissue penetration, and sensitive biosensors. Despite their tremendous success, UCNMs face a number of problems: (1) The stability of luminescence efficiency after surface modification: the oil-soluble molecules will be changed to increase biocompatibility during surface modification. Nonetheless, the capacities of UCNMs to disperse in oil and water differ. Fluorescence quenching and reduced upconversion efficiency are common after surface modifications. (2) The biological toxicity of UCNMs: several studies have demonstrated that suitable adjustments of the chemical composition, particle size distribution, and surface modification may significantly enhance the biocompatibility of UCNMs for biomedical applications. However, no investigations have been performed to assess their long-term toxicity, including possible immunological responses and mutagenic effects. (3) Technical deficiencies in clinical trials: UCNM-based phototherapy has not been tested in humans, owing to concerns about biosafety and therapeutic efficacy. There is still a long distance from laboratory animals to human-level technical standard revisions. In summary, UCNMs offer a fantastic opportunity to achieve precision medicine. We anticipate that UCNMs will become more competitive in biological applications following stable surface modification, minimized toxicity, and clinical studies.

## Figures and Tables

**Figure 1 nanomaterials-12-03470-f001:**
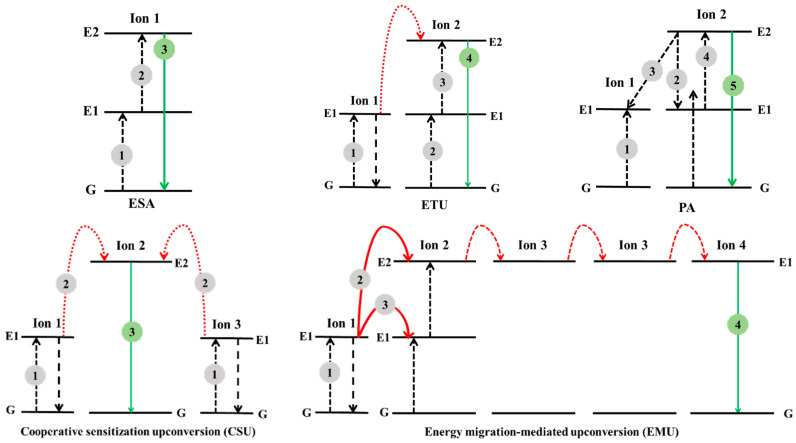
Overview of upconversion mechanisms.

**Figure 2 nanomaterials-12-03470-f002:**
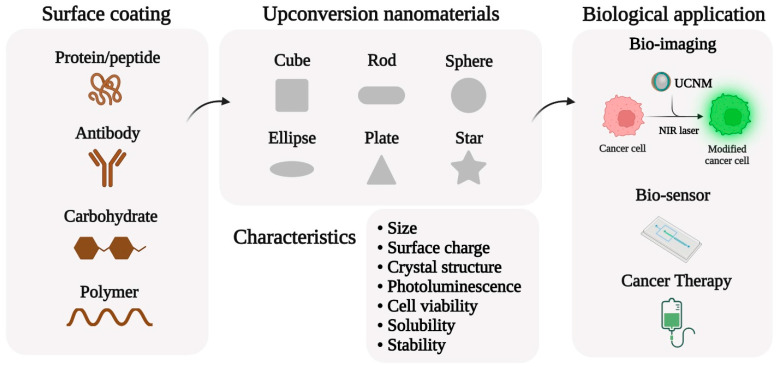
Various properties of upconversion nanomaterials and biological applications.

**Figure 3 nanomaterials-12-03470-f003:**
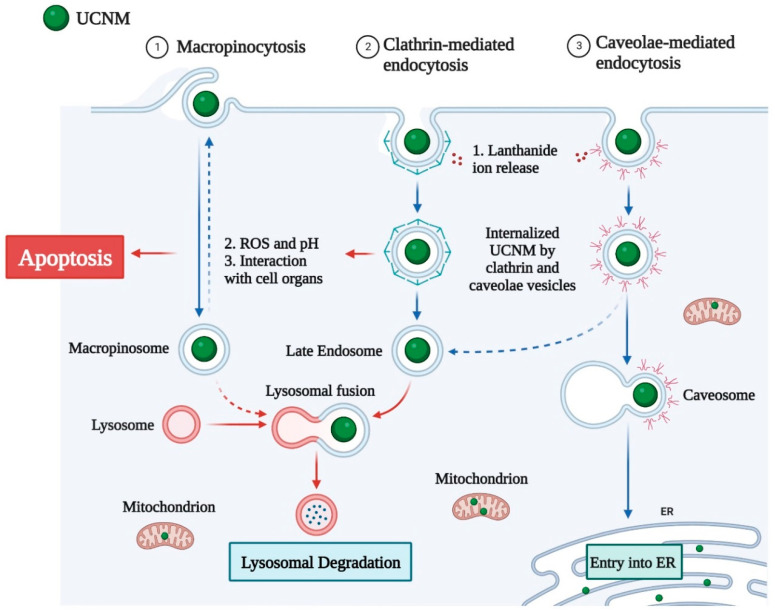
Cellular uptake and intracellular mechanism of upconversion nanomaterials.

**Table 1 nanomaterials-12-03470-t001:** Cytotoxicity of upconversion nanomaterials in vitro.

Nanoparticles	Size (nm)	Shape	Surface Ligand	Cell Viability	Cell	Conc.	Ref.
NaYF_4_:Yb^3+^/Er^3+^	50	Hexagonal	SiO_2_	n/a	In vivo: mice	n/a	[40]
NaYF_4_:Yb^3+^/Tm^3+^	n/a	n/a	TiO_2_	n/a	HepG2-CCK-8 assay	n/a	[41]
NaGdF_4_/GdF_3_:Yb^3+^,Er^3+^	30	Hexagonal	PEG	90%	HeLa, MCF-7, A549	62.5, 125, 250, 500 μg/mL	[42]
NaGdF_4_/GdF_3_:Yb^3+^,Er^3+^	30	Hexagonal	OA	63%	HeLa, MCF-7, A549	62.5, 125, 250, 500 μg/mL	[42]
LiLuF_4_:Yb,Er@nLiGdF	8.5 (core) 24.2 ± 1.25	n/a	mSiO_2_	92.3%	MCF-7	50, 100, 200, 400, 800 μg/mL	[43]
NaYF_4_:Yb^3+^/Er^3+^	47 ± 1 5 ± 2	Hexagonal	Thin SiO_2_ Thick SiO_2_	51 ± 5%, 110 ± 12& 75 ± 6% 95 ± 14%	RAW 264.7	12.5 and 200 μg/mL	[44]
NaY_0.78_Yb_0.2_Er_0.02_F_4_	93 ± 8	Hexagonal	PEI PEI@Dx PEI@DxS PEI@SCh PEI@PAMA PEI@PAMA@Dx PEI@PAMA@Ch	63%	murine bone marrow mesenchymal stem	1–25 μg/mL during 1–2 days	[45]
NaYF_4_:Yb,Er@NaYF_4_:Nd@NaYF_4_	35	Irregular	Fe(OH)_3_				[46]
Y_2_O_3_:Er^3+^/Yb^3+^	70 ± 10	n/a	Folic acid	>80%	HeLa, MDA-MD-231, MCF-7	1 μg/mL	[47]
NaYF_4_:Yb^3+^,Er^3+^	22 ± 1	Hexagonal	AMPA APTES DHCA MAEP	>80%	CHO-K1	Low 10 μg/mL High 100 μg/mL	[48]
NaLuGdF_4_:Yb^3+^/Er^3+^ (Tm^3+^)	~80	n/a	Malonic acid	>80%	HeLa cell	0–600 μg/mL	[49]
NaGdF_4_:Yb^3+^, Er^3+^	3.5 ± 0.4 16.6 ± 1.5 249 ± 59	Hexagonal	NOBF_4_	IC_50_ = 0.81 ± 0.06 μg/mL IC_50_ = 1.33 ± 0.07 μg/mL IC_50_ = 1.58 ± 0.05 μg/mL	RAW 264.7 J774A.1	0.1–50 μg/mL	[50]
NaYF_4_:Yb^3+^,Er^3+^	19.9 ± 1.1 19.8 ± 1.3 19.7 ± 1.8 20.0 ± 1.5	Hexagonal	Citrate AA EDTMP PMAO	>80%	Human HaCaT keratinocyte cells	12.5–200 μg/mL	[51]
NaYF_4_:Yb^3+^,Er^3+^	26.1 ± 1.9 47.8 ± 1.9	Hexagonal	Cucurbit[n]urils (CB[n])	>50%	RAW 264.7 HeLa cell	50 μg/mL	[52]

Abbreviations: AMPA, (aminomethyl)phosphonic; APTES, 3-aminopropyl)triethoxysilane; DHCA, 3,4-dihydrocinnamic acid; MAEP, poly(monoacryloxyethyl phosphate; AA, alendronate, EDTMP, ethylenediamine tetra(methylene phosphonate); PMAO, poly(maleic anhydride-alt-1-octadecene).

## Data Availability

Not applicable.

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
