# Peer review of "Upconversion Nanomaterials in Bioimaging and Biosensor Applications and Their Biological Response"

_nanomaterials, 2022, doi:10.3390/nano12193470_

Round 1

Reviewer 1 Report

1. The text in some pictures is too small, which makes it inconvenient to read. It is recommended to change the font size to make the pictures more intuitive and easy to read.

2. The introduction of the application of upconversion nanomaterials is too simple. I believe that some specific cases will be cited to make it easier for readers to understand.

Author Response

Comment 1. The text in some pictures is too small, which makes it inconvenient to read. It is recommended to change the font size to make the pictures more intuitive and easy to read.

Response: We are sorry about the text in the pictures is too small. We updated new images with better quality.

Comment 2. The introduction of the application of upconversion nanomaterials is too simple. I believe that some specific cases will be cited to make it easier for readers to understand.

Response: Thank to the reviewer’s comment. We have discussed more applications of UCNM in the introduction section with cited specific cases in the manuscript.

Page No. 1, lines No. 45-54.

Reviewer 2 Report

The manuscript forays into the theory, still outstanding issues, and possible applications of upconversion nanomaterials (UCNMs) in precision medicine.

Table 1 represents an interesting collection of information regarding some of the already investigated applications of UCNMs.

Issues to be addressed by authors

1. Even if the final version will not include numbered lines, the fragmented pages numbering and lines numbering after the insertion position of Table 1 makes the manuscript difficult to read.

2. Please verify reference [31]. Enumeration in lines 185-186 cannot be attributed to reference [31], but, perhaps, to reference [97].

3. Kout should be written by using subscript fonts: Kout.

4. In reference [96] description (last page of your manuscript), please use "pH" instead of "PH".

5. As a personal opinion (not necessarily to be considered by the authors), the biological toxicity of UCNMs, which is well mentioned in section 6, should be mentioned in the abstract also.

6. Please review the references numbering and their appropriateness and consistency with the text of the manuscript.

Author Response

Comments from Referee 2:

The manuscript forays into the theory, still outstanding issues, and possible applications of upconversion nanomaterials (UCNMs) in precision medicine.

Table 1 represents an interesting collection of information regarding some of the already investigated applications of UCNMs.

Response: We would like to thank the reviewer’s comments.

Comment 1: Even if the final version will not include numbered lines, the fragmented pages numbering and lines numbering after the insertion position of Table 1 makes the manuscript difficult to read.

Response: We are sorry for the inconvenience. Line numbering included in the whole manuscript from 285 to 682

Comment 2: Please verify reference [31]. Enumeration in lines 185-186 cannot be attributed to reference [31], but, perhaps, to reference [97].

Response: Reference of enumeration in lines 185-186 verified and cite [97] 

Page no. 3, line no 150.

Size is critical in biological-NP interactions. The biological-NP-related mechanisms include pinocytosis, phagocytosis, clathrin-mediated endocytosis, caveolae-mediated endocytosis, clathrin- and caveolae-independent endocytosis, and micropinocytosis [40].

Comment 3: Kout should be written by using subscript fonts: Kout.

Response: Thank you for pointing this out. We agree with this comment and revise in line number 249.

Page no. 7, line no 296

The originality of these studies is that the Kout of the nanoprobes accumulated in the tumor resulted in a 4-fold accumulation of probes, offering stable cancer retention for up to 5 h by organizing the probes into larger assembled clusters.

Comment 4: In reference [96] description (last page of your manuscript), please use "pH" instead of "PH".

Response: Thank you for pointing this out.

Tsai, E.S.; Joud, F.; Wiesholler, L.M.; Hirsch, T.; Hall, E.A.H. Upconversion Nanoparticles as Intracellular pH Messengers. Anal. Bioanal. Chem. 2020, 412, 6567–6581, doi:10.1007/s00216-020-02768-5.

Comment 5: As a personal opinion (not necessarily to be considered by the authors), the biological toxicity of UCNMs, which is well mentioned in section 6, should be mentioned in the abstract also.

Response: Thank you for your comment. We included the biological toxicity of UCNMs in the abstract.

Page no. 1, line no 19-21.

Additionally, biological toxicity of UCNMs were explained and summarized with intracellular association mechanisms. Finally, the prospects and future challenges of UCNMs at the clinical level in biological applications are described with summary of opportunity of UCNM’s biological as well as clinical application. 

Comment 6: Please review the references numbering and their appropriateness and consistency with the text of the manuscript.

Response: Thank you for your comment.

Reviewer 3 Report

In this short review, authors tried to depict the development of upconversion nanomaterials, its features and potential application in bioimaging and biosensor. The content is helpful and important for academic field and upconversion nanomaterials.

1. In introduction section, the tedious description about the preparation methods should be removed.

2. As a review, the corresponding literature must be cited to support description. For instance, the phrase of For example, synthetic organic dyes are rapidly photobleached, rendering themselves unsuitable for longitudinal tracking tests. Common binary quantum dots contain highly toxic components that may hamper their clinical application.’

Author Response

Comments from Referee 3:

In this short review, authors tried to depict the development of upconversion nanomaterials, its features and potential application in bioimaging and biosensor. The content is helpful and important for academic field and upconversion nanomaterials.

Response: We would like to thank reviewer’s comments.

Comment 1: In introduction section, the tedious description about the preparation methods should be removed.

Response: Thank you for your comment, section named “synthesis of upconversion nanomaterials” is minimized and gives a short introduction to the synthesis of UCNMs.

Comment 2: As a review, the corresponding literature must be cited to support description. For instance, the phrase of ‘For example, synthetic organic dyes are rapidly photobleached, rendering themselves unsuitable for longitudinal tracking tests. Common binary quantum dots contain highly toxic components that may hamper their clinical application.’

Response: Thank you for your comment, the corresponding literature cited in the test to support the description.

Page No 6, line no 271-280.

When excited by near-infrared light (808-980 nm), UCNMs display distinct narrow photoluminescence with higher energy (visible light). Because of their distinctive upconversion luminescence (UCL), UCNMs have the potential to be bioimaging probes with appealing properties, such as minimal auto-fluorescence from biological materials and deep penetration depth [68,69]. Consequently, UCNMs have emerged as innovative small-animal imaging agents. Traditional fluorescent materials have limitations that make them unsuitable for high-contrast in vivo imaging [70]. For example, synthetic organic dyes are rapidly photobleached, rendering themselves unsuitable for longitudinal tracking tests [71]. Common binary quantum dots contain highly toxic components that may hamper their clinical application.